# The Interplay between Housing Environmental Attributes and Design Exposures and Psychoneuroimmunology Profile—An Exploratory Review and Analysis Paper in the Cancer Survivors’ Mental Health Morbidity Context

**DOI:** 10.3390/ijerph182010891

**Published:** 2021-10-16

**Authors:** Eva Hernandez-Garcia, Evangelia Chrysikou, Anastasia Z. Kalea

**Affiliations:** 1The Bartlett Real Estate Institute, The Bartlett School of Sustainable Construction, University College London, London WC1E 6BT, UK; e.chrysikou@ucl.ac.uk; 2Clinic of Social and Family Medicine, Department of Social Medicine, University of Crete, 700 13 Heraklion, Greece; 3Division of Medicine, University College London, London WC1E 6JF, UK; a.kalea@ucl.ac.uk; 4Institute of Cardiovascular Science, University College London, London WC1E 6HX, UK

**Keywords:** housing, psychoneuroimmunology, inflammation biomarkers, household environmental health, non-pharmacological interventions, therapeutic architecture, supportive domestic spaces, mental health comorbidities, cancer survivorship, design for wellbeing

## Abstract

Adult cancer survivors have an increased prevalence of mental health comorbidities and other adverse late-effects interdependent with mental illness outcomes compared with the general population. Coronavirus Disease 2019 (COVID-19) heralds an era of renewed call for actions to identify sustainable modalities to facilitate the constructs of cancer survivorship care and health care delivery through physiological supportive domestic spaces. Building on the concept of therapeutic architecture, psychoneuroimmunology (PNI) indicators—with the central role in low-grade systemic inflammation—are associated with major psychiatric disorders and late effects of post-cancer treatment. Immune disturbances might mediate the effects of environmental determinants on behaviour and mental disorders. Whilst attention is paid to the non-objective measurements for examining the home environmental domains and mental health outcomes, little is gathered about the multidimensional effects on physiological responses. This exploratory review presents a first analysis of how addressing the PNI outcomes serves as a catalyst for therapeutic housing research. We argue the crucial component of housing in supporting the sustainable primary care and public health-based cancer survivorship care model, particularly in the psychopathology context. Ultimately, we illustrate a series of interventions aiming at how housing environmental attributes can trigger PNI profile changes and discuss the potential implications in the non-pharmacological treatment of cancer survivors and patients with mental morbidities.

## 1. Introduction

Since the onset of COVID-19 outbreak, global attention has largely focused on the impact of the pandemic and the mental health burden on the general population, and particularly on vulnerable people with pre-existing mental illness and other non-infectious chronic comorbidities [1,2,3]. Access to healthcare was in several cases restricted, either because many primary care settings closed at the beginning of the pandemic, or because several outpatient services closed, and inpatient wards had to undergo urgent adjustments, as a result of COVID-19 preparedness plans. In short, for some people, the lockdown policies meant that people who otherwise would have sought help either from their GP’s or mental health services stayed at home [4]. While public health emergency measures including stay-at-home policies have become critical to mitigate the spread of infectious disease [5,6], the same policies may have adverse effects for people with mental illness as the timing of intervention is critical. Whereas the COVID-19 home confinement undoubtedly involved short- and long-term adverse mental health and emotional well-being consequences on residents [7,8,9,10,11], some people demonstrated resilience over the following months against this challenging disruptive event [12,13,14,15,16]. This variance of mental health outcomes during the COVID-19 crisis has been directly associated with residential environment and housing-related characteristics. Populations having access to domestic environments demonstrating high-quality traits compared with those people living in deprived urban areas, without access to residential public green spaces, lacking private natural spaces and/or frequency of viewing nature elements from home, housing interior design with poor natural lighting, and lower level of perceived indoor air quality had a protective effect against detrimental mental wellbeing outcomes [17,18,19,20,21,22].

The COVID-19 situation is a particularly troublesome experience for people living with and beyond cancer, especially those immunocompromised adult patients due to their higher comorbidity burden [23]. With substantial delays in diagnosis and treatments, lower capacity for surgery, and reductions in routine supportive services [24], the crisis event may have potentially led to worse overall patient survival outcomes [25]. Although evidence is still emerging, the psychosocial distress and physical wellbeing of cancer survivors was wide-ranging, exacerbated, and strongly stressed the need for drastic changes to cancer survivorship care in terms of frequency, type and mode of care delivery [26]. The main themes addressed in neighbourhood environment-related cancer survivorship care research are focused across survival disparities, primary care, and weight management, followed by quality of life and environmental exposure [27]. Scoping evidence shows how built environment features regarding urban design, land use, spatial accessibility and housing socioeconomic status are related to risk factors of cancer such as screening adherence, residential mobility, diet patterns, contaminated water and air quality [28,29]. However, the correlate and causal relationships of built environment, namely housing design and environmental characteristics, cancer survival outcomes and care, and psychological wellbeing and mental comorbidities are understudied [27,29]. The individual’s systemic chronic inflammatory burden has been postulated as a potential driver for multiple pathologies, including psychiatric diseases, and as a clinical predictor of multimorbidity [30]. In this context, the modulation of psychoneuroimmunology (PNI) axis and its relevant physiological information from circulating mediators may be a key catalyst between psychopathology and housing characteristics and expand the in-depth understanding of the domestic recovery spaces for mental state.

In this paper, we present a line of evidence-supported interdisciplinary arguments tackling the unique complexity of mental health burden in the cancer survivorship, while exploring the COVID-19 related disruptions on new forms of care delivery and of domestic spaces to manage the psychological and psychiatric dimensions. Our main goal is to highlight the crucial role that a PNI-oriented approach could play in boosting the constructs of physiologically supportive housing, especially for mental health comorbidities in cancer survivors. We argue how this novel approach may enable opportunities to promote transformative changes, placing housing as an essential healthcare component for the emerging sustainable primary care, and public health integration into the new models of primary care-based cancer survivorship care.

From non-pharmacological treatment and clinical implications perspectives, we ultimately conducted a systematic-type search and brief analysis of intervention trials aiming to address how the exposures to housing conditions and the quality of the indoor environment directly interfere with PNI indicators. We look at the current empirical evidence and evaluate whether this exploratory review could serve as a precursor for a comprehensive systematic review and meta-analysis.

## 2. The Complexity of Mental Health and Comorbidities in Cancer Survivorship

Clinically significant psychopathology symptoms are detected throughout the course of cancer survivorship up to several years post-diagnosis and treatment have been well-documented, even before the COVID-19 pandemic [31]. Compared with the general population, adult cancer survivors have an increased prevalence of mental health comorbidities including major depressive and anxiety episodes, severe psychological distress, bipolar disorders, obsessive-compulsive disorders, neurocognitive dysfunction and suicidal outcomes [32,33,34,35,36,37,38], as well as other adverse late-effects interdependent with mental illness outcomes such as post-cancer chronic pain, fatigue and sleep disturbances [39,40,41,42,43]. Besides the pre-existing medical conditions at the time of cancer diagnosis [44], many of these chronic condition and multimorbidity clusters are identified following cancer [45,46]. Multimorbidity is more likely in cancer survivors, between 46% and 69% reported two or more comorbid conditions, which increase over time due to population ageing [47,48,49,50,51]. This value could increase depending on the type of cancer and specific age, and multimorbidity clusters up to 84% for long-term breast cancer survivors [52]. Although no gold standard for defining and measuring the level of multimorbidity is established with 33 unique instruments, multimorbid patients are commonly defined from a public health and epidemiology perspective as having two or more currently long-term medical conditions to a range of 4–147 different conditions [53,54,55,56]. However, using common measures, including the disease counts and weighted indices tools may not well-characterise the complex profile of cancer survivors. From a more holistic perspective, the multimorbidity term drawn primarily means any combination of chronic illness with at least one other medical condition (either chronic or acute) or biopsychosocial determinant or somatic risk factor [57], as this encompasses better relevant clinical characteristics in the trajectory of cancer survivorship and is particularly suited to long-term care and primary healthcare [58,59,60].

In this context, polypharmacy requiring the use of psychotropic drugs among cancer survivors [49,61,62,63], as well as other coexisting medical conditions, increases the demand for healthcare and mental health services [63,64,65]. Across several European countries, patients with multimorbidity have up to 3.5 times as many primary care consultations as individuals without any diagnosed chronic condition [66,67,68]. Interestingly, these medical doctor visits are associated with the cluster composition of morbidity, namely the combination of mental and physical conditions [69,70,71,72,73,74]. The proportion of multimorbid patients who suffer from a physical–mental comorbidity is higher among women, younger age groups (18–44 years), and individuals with lower socioeconomic status [75,76]. Looking at age-stratified clusters of multimorbid patients, clusters with the most general practice (GP) contact in a year and regular medications comprised depression, anxiety and pain in people aged under 65 years old and the co-morbid depression, coronary heart disease and pain in people aged 65–84 [77]. Cancer survivors suffering from depression comorbid with two other chronic morbidities report having a decremented mental health-related quality of life [48,78]. Furthermore, specific multimorbidity clusters are associated with poor cancer survival outcomes [79]. For instance, colorectal and gastrointestinal (GI) cancer patients with co-morbid rheumatologic disease and diabetes, and clustered with multiple sclerosis and GI disorders, respectively, could have higher short-time mortality risk compared to those without these multimorbid cluster profiles [80,81]. Breast cancer survivors presenting musculoskeletal and cardiovascular disorders have an increased risk of death than those with other multimorbidity clusters [52]. Overall, the clinical trajectory of multimorbid people in clusters characterised by neuropsychiatric diseases show higher mortality rates over time [82].

Multimorbidity clusters with mental health conditions are recognised as the main impact on GP contact and associated with worse quality of life and survival outcomes. Much has been published over the past decades about the challenges and potential solutions to provide high-quality supportive cancer survivorship care, particularly to overcome unmet supportive care needs in their complex psychosocial vulnerabilities and to move towards an optimal integration of primary care and community-based cancer rehabilitation programmes [83,84,85,86,87,88,89,90,91]. In this context, interventions to manage multimorbid patients and improve mental health-related outcomes may be more effective if these are focused on the specific risk factor, such as depression [92]. Identifying profiles of multimorbidity clusters among cancer survivors may highlight major targets of risk factors to improve health outcomes, as well as facilitate a patient-centred holistic cancer survivorship care.

## 3. COVID-19 Disruptive Changes Adding Hardship to Psychological Stressors

The implications of the COVD-19 pandemic have led to further stressors on the physical, psychosocial, and healthcare delivery concerns of cancer survivors. Research conducted in the COVID-19 outbreak context reports that the pandemic provided a new multifaceted source of mental health-related problems for cancer survivors because of several influences such as cancer service disruptions, social isolation, financial stress to pay medication and fear of getting COVID-19 infection [93,94,95,96,97]. For instance, breast cancer survivors returning to rehabilitation services after the COVID-19 interruption reported lower emotional distress levels than during the time of services suspension [98].

Cancer survivorship is an important risk factor for COVID-19-related severe health outcomes, since survivors generally present a combination of comorbidities, older age ranges [99,100,101,102] and potentially weakened immune systems, associated with immunosenescence and low-grade systemic inflammation [103,104,105,106]. Moreover, cancer survivors have been concerned about delays and disruptions in healthcare services and uncertainties related to their future treatment [96]. Reductions and delays of routine activities in any stage of cancer care services—including screening services, treatment, surgery, biopsies, physical therapy and rehabilitation, outpatient visits, relocation of care, palliative care—became a global normality following the COVID-19 pandemic onset across Europe, North America and the Asia–Pacific region [24]. After the introduction of population-wide restrictions in different countries, the majority of multimorbid patients were either disengaged from outpatient services or routine chronic care and cancer rehabilitation services from their primary care providers or based in the community since the pandemic [107,108,109,110,111]. In the United Kingdom (UK), one of the largest reductions in primary care contacts was reported for depression, besides other mental conditions including anxiety disorders and severe mental illness, during the April 2020 lockdown [110,112]. In this period, the rate of referral to mental health services was 75% less than the expected for the time of year, with the largest reductions for people living in deprived areas [112]—a population group who have a more deteriorated mental health status [113,114]. However, only few cancer survivors have been receiving professional outpatient mental health care services prior to COVID-19, mostly for younger adults [115].

Built environment attributes impact on resident’s mental health status. During COVID-19, the role of the residential built environment and housing on resident’s mental health has become apparent as people had to made significant changes in the way they interacted, the types of buildings they were allowed to access and the quality of the environment where they had to self-isolate [116]. Increased risk of mood disorders has been associated to the exposure to dwelling characteristics, home environment attributes and the affordances that home would allow during COVID-19 [117]. Although there are limited empirical studies, the population reported physical or mental health problems following the imposition of COVID-19 lockdown measures because of inadequate housing conditions, including overcrowding, lack of space, poor views, poor indoor environmental and design quality, such as minimal natural lighting and poor acoustics, and no access to external open spaces like balconies, terraces or domestic gardens [18,117,118,119,120,121].

### 3.1. Substantial Changes of Ambient and Household Environmental Quality during COVID-19 Lockdown

While housing has become a primary protection and mitigation strategy against COVID-19, the physiological and pathological response to home environments and its associated household environmental quality domains either supporting or undermining mental health was initially negligible.

Lockdown dates over different countries resulted in drastic reductions in global-level air pollutants compared to those expected for the time of year; however, variations in air quality were dependent on the specific implemented measures and timing of lockdowns [122,123,124]. Overall, whereas ambient air pollution levels decreased in most of the forms, an increase of ozone concentrations were reported in several regions of Europe, including Spain [125], France, Italy [126], Netherlands [127], and the UK [128,129], as well as China [126,130], South and Southeast Asia [131,132], Canada [133], the United States (US) [134,135] and Mexico [136], despite different lockdown measures being implemented. Residential environments remained a significant risk source to indoor air emissions [137,138]. It was observed that indoor particulate matter levels had an almost 3-fold increase in residential settings during and post-lockdown periods [138]. After the imposition of stringent stay-at-home policy, household volatile organic compounds (VOCs) concentrations reached the highest values compared to other periods, as residents spent longer time in their houses while activities outside home were restarted gradually during the re-opening stage [139]. The anthropogenic VOCs emissions-related residential environments during the lockdown may have had an impact on the total ozone formation potential in different areas [140]. Besides household air pollution deriving mainly from furniture, building materials, cooking, cleaning, lighting and heating activities, engineering system defects or presence of humidity, the increased use of disinfectant products, and lack of suitable ventilation routines and mechanisms rose the indoor pollutant concentrations of dwellings during the lockdowns, constituting those unhealthy and unsustainable environments [122,141,142,143]. Poor household air quality could exacerbate the mental health morbidity and cancer survival outcomes, as evidence has linked the exposure to residence place-related air pollution with several mental disorders, such as depression, anxiety and suicide risk [144,145], and cardiopulmonary mortality in cancer survivors, especially those who received chemotherapy or radiation treatments [146]. In some households, those risks could be mitigated by several means. For example, exposure to the increased cooking activity at home during the lockdown was mitigated through the use of room air cleaners [147].

### 3.2. Buffering Effect of Housing Conditions on Resident’s Mental Health during COVID-19 Lockdown

Residents with access to some environmental affordances at the time of lockdowns experienced better mental health status during COVID-19 (Table 1). Elements included houseplants, home garden access and usage and visibility of greenery and/or blue spaces from the windows [18,148,149,150,151,152], as well as indoor soundscape prominently coming from nature [153] as mechanical noise like traffic noise reduced [154]. Across several European countries, including France, Germany, Italy, Portugal, Spain, and the UK, as well as New Zealand, US and Mexico, people under lockdown at homes with accessible outdoor spaces and views of nature showed lower symptoms of depression and anxiety [18]. In addition, having different forms of vegetation within homes positively supported the population’s mood from several European countries, the US and South America during confinement [152].

During Spain’s stricter lockdown, people who maintained or increased the exposure to houseplants and community private green spaces presented remarkably lower levels of stress and somatization, respectively [19]. Moreover, home characteristics related to types of elements for outdoor accessibility (balcony, garden/patio, shared or public spaces) and types of views from the home (natural, limited/urban, mixed views) was associated with the likelihood of exhibiting symptoms of mental disorders [18]. For instance, Spanish people residing in dwellings with nature features in their views or access to a private garden/patio—compared with access to balcony space and public or shared outdoor spaces—reduced the likelihood of clinically important symptoms of depression and overall mental health [18]. Higher levels of visibility of houseplants and/or natural greenery from a window, terrace or balcony were associated with reduced depressive and anxiety symptoms among Bulgarian young adults [148]. The presence of a greater amount of plants, green views from windows and availability of access to private natural spaces were strongly associated with improved psychological health outcomes—anxiety, moodiness and sleep disturbance—in Italian dwellers during the COVID-19 home confinement [155]. In Singapore, healthy adults with lower nature exposure during the stay-at-home order reported an increase in the severity of depressive symptoms post-COVID-19-related lockdown [156].

Moreover, in Portugal, the contact with public natural spaces was significantly and negatively associated with stress levels, as well as views of nature with lower levels of psychological distress, somatization and stress [19]. When the place of residence had surrounding greenness, Swedish people in contact with nature also demonstrated better mental health and less depression, cognitive stress, and anxiety symptoms [157]. In the same line, Japanese residents with access to nearby neighbourhood greenness show improved mental health outcomes, but surprisingly those living at home environments with green window views had greater decreased levels of depression and anxiety [149]. Interestingly, differences in the effects of indoor/outdoor green features on mental health has been reported; while the access to private natural spaces resulted in an improvement in a myriad of psychological outcomes, the exposure to neighbourhood green environment could have a moderately restorative effect [155]. In this line, Scottish older adults who spent more time in domestic gardens during the lockdown experienced better mental health and sleep quality compared with pre-lockdown [150]. By contrast, individuals in England residing close to a neighbourhood-scale greenery within a 250 m spatial buffer had higher levels of mental wellbeing and better coped with the COVID-19 lockdown compared to a proximal land-cover greenness such as private gardens or green spaces out of the neighbourhood scale [158]. The association between both home and public natural spaces and psychopathological distress improvements may be linked not only by the presence of vegetation, but also with certain activities carried out in these areas like exercising, gardening, enjoying fresh air and sunlight or engaging with other people [151,152,158,159].

Moreover, it has been suggested that beyond the quantity of nature, the quality of green spaces including natural features, and the frequency and duration of viewing or visiting, and the accessibility to green spaces—inequitably distributed—could be a determinant in the psychophysiological responses [19,149,150,156].

**Table 1 ijerph-18-10891-t001:** An overview of the main effects of exposure to housing environmental conditions on residents’ mental health outcomes during COVID-19-related restrictions or confinement.

Residents’ Location [Reference]	Home Environmental Attributes	Mental Health Outcomes
Portugal, Spain [18,19]; Italy [18,155]; France, Germany, United Kingdom, New Zealand, United States, Mexico [18]; Bulgaria [148]; Scotland [150]; Sweden [157]; Japan [149]	Accessibility to outdoor spaces and types of elements (private domestic garden/patio compared to balcony and shared/public spaces). Contact with public natural spaces nearby the place of residence.	Lower likelihood of symptoms of depression and anxiety, and mental disorders and sleep disturbances. Decreased levels of depression, anxiety, somatization, cognitive stress, and/or psychological distress.
Portugal [18,19]; Italy [18,155]; France, Germany, Spain, United Kingdom, New Zealand, United States, Mexico [18]; Bulgaria [148]; Japan [149]	Visibility of natural greenery from window, terrace, or balcony.	Decreased levels of depression, anxiety, somatization, cognitive stress, psychological distress, and/or sleep disturbances.
Spain [19,152]; Chile, Colombia, Brazil, Argentina, Mexico, United States, United Kingdom, France, Italy, Germany, Greece [152]	Having higher levels of and/or different forms of vegetation within homes.	Reduced risk of mood disturbances and lower levels of stress and somatization.
Italy [152,155]; Spain [22,152]; Chile, Colombia, Brazil, Argentina, Mexico, United States, United Kingdom, France, Germany, Greece [152]	Adequate levels of natural lighting in indoor spaces (i.e., homes with more surface area of window openings).	Reductions in moodiness, sleep disturbances and negative emotions.
Bulgaria [153]	Nature-generated soundscape patterns. Adequate human-generated sound levels (i.e., dwellings with soundproof windows minoring neighborhood or mechanical noise).	Increases in the restorative capacity and improved overall well-being.

### 3.3. Housing Design and Constructive Characteristics Impact on Environmental Conditions

Housing typology was a key aspect associated with the exposure and usage frequency of certain spaces with nature elements and/or neighbourhood green infrastructure (Table 1). For instance, people residing in urban single-family houses and apartments/top-floor apartments during lockdown were associated with lower odds of maintaining or increasing the use of public natural spaces and views of nature [19]. In addition, people living in small size houses and with lower levels of natural lighting, with no or minimal vegetation, were more prevalent to have negative emotions [152]. More specifically, people teleworking for extended periods in homes without a horizontal surface area of window openings and accessibility to patios or terraces had higher incidence of suffering from certain disorders related to circadian rhythm, due to inadequate levels of natural lighting [22]. In this context, higher levels of sunlight in the home environments have been associated with reductions in moodiness and sleep disturbance among dwellers in lockdown [155].

After declared lockdowns, most large cities including Barcelona, Madrid (Spain) and Bochum (Germany) (who had a strict lockdown), and London (UK), and Stockholm (Sweden) (who had laxer mobility restrictions), observed reductions of traffic, and industrial/commercial and human activities-related noise pollution levels of an average of 6.3 dB, 4.2 dB, 5.1 dB, 5.4 dB and 2.7 dB, respectively, compared with the pre-lockdown period; although, importantly, variations were dependent on the urban and residential area types, pre-existing sound sources and imposed measures [159,160,161,162,163]. Soundscape patterns also changed during the COVID-19 outbreak [162]. Human-generated sound levels such as talking and walking, as well as nature sounds such as birdsong, through open windows increased within home environments [154]. Still, greater exposure to mechanical noise during home confinement contributed consistently to a resident’s worse self-reported health, whereas nature sounds correlated with greater restorative capacity of the home; however, those dwellers having soundproof windows installed in their homes, and therefore more acoustic comfort, reported better overall wellbeing independently associated [153]. In contrast, where construction/public work and neighbourhood noise was reported to become an annoyance for residents, housing characteristics were an important factor of indoor acoustic environment and complaints during lockdown [164,165].

## 4. The Interconnectedness between Housing, Public Health and Primary Healthcare

### 4.1. New Models of Primary Care-Based Comprehensive Cancer Survivorship Care

COVID-19 heralds an era for renewed call for actions to identify sustainable methods and modalities to adapt health care service delivery to cancer survivors [111,166,167,168] and improve housing conditions and indoor environment quality for health recovery [169,170]. According to the Cancer Survivorship Care Quality (CSCQ) Framework, the interrelated key domains of optimal survivorship care and health care delivery, classified in this review into three blocks, include (a) surveillance and management of physical and psychosocial effects and chronic medical conditions, (b) prevention and surveillance of recurrences and new cancers, as well as (c) health promotion and disease prevention [171]. Importantly, these constructs serve as references for application in research initiatives by using disease-based indicators and appropriate outcome measures to capture the care [171]. An evidence map identified multiple model types for delivering survivorship care which are highly individualised to the institution where cancer survivors are based, with great heterogeneity in the model components (i.e., surveillance, prevention, intervention, coordination), care providers (i.e., physician-, nurse-, practitioner-, care team-led) and target outcomes [172]. In the last six years, cancer survivorship care is moving towards models centred in primary health care where survivors are viewed in the integrality of their complex morbidities, more likely to be in line with CSCQ Framework, with referrals to specialists as appropriate, and an emphasis on preventive medicine [173,174]. Primary care is now characterised by multi-professional teams and practices are increasingly collaborating with networks and/or federations to facilitate cost-effective care on a large scale [175]. In those regions with strong primary care foundations, cancer survivorship primary care models have proven effective for patients. For instance, over a third of patients from the John Hopkins Primary Care for Cancer Survivors Clinic programme (Baltimore, US) received speciality referrals to physical therapy, physical activity, clinical nutrition support, psychosocial services and local support organisations to manage their emotional needs, long-term mental and physical effects—namely fatigue and weight change—comorbidities and fear of recurrence [173]. The sustainable Clinical Oncology Society of Australia (COSA) model recognises primary care as the adequate place to provide preventive and supportive care for the post-treatment phase of cancer survivors, and also encompasses specialist referrals to a wide range of health professional services and emphasises the availability of local services [176]. Interventions based on this type of new model of care via the Victorian Cancer Survivorship Program (second phase, 2016–2019) showed to be acceptable, appropriate, and effective for Australian cancer survivors, but also added growing evidence to advance around post-treatment care [177]. The systemic, multidisciplinary and shared approach of the COSA model not only is supporting healthy lifestyle behaviours and primary and secondary prevention while managing late and long-term side effects and comorbidities, but also is looking for minimising unnecessary healthcare service use while incorporating stratified pathways of care centred on survivor’s risk factors [178]. In England, the National Health Service (NHS) trust has implemented the personalised, risk-stratified follow-up care approach since 2015, which assessed the cancer survivors’ risk of recurrence, new cancers and late outcomes [179].

However, primary health settings in most high-income regions have not begun to enforce this comprehensive survivorship care model, neither are they looking to transition their own follow-up systems. From Australia, USA, UK and the Netherlands, primary care practices and the Cancer in Primary Care International Network (Ca-PRI) annual scientific meeting highlighted that the absence of distinction of cancer survivorship as a clinical category, lack of primary care provider’s skills and training, limited guidance—low specificity and consistency—for long-term follow-up care and inadequate information and referral systems for implementing population-level interventions and support survivorship care could complicate the clinical care into primary care [180,181,182].

### 4.2. Housing as an Essential Component for Sustainable Primary Care and Public Health Integration

Establishing the key constructs of the CSCQ enforces the integration of primary health care and public health sectors. In many European and North American countries, a growing number of public health functions are provided as primary care, such as interventions applying a population perspective to clinical practice and supporting health promotion and disease prevention, as well as undertaking community health problems [183,184]. For instance, public health practitioners and GP practices in Liverpool (UK)—organised into 18 neighbourhoods—have collaborated to develop a neighbourhood-associated health profile using electronic health records (EHR) in order to inform primary care practice [185]. In this context, the London borough-local Primary Care Trusts having higher densities of street trees has been associated with 1.4 fewer anti-depressant prescriptions per 1000 population [186]. Routinely collected data in primary care can provide robust evidence to inform the optimal allocation of resources, including those related to residential settings and patients with mental disorders [187]. Since the global commitment of the 2018 Astana Declaration pledged for the integrated delivery of sustainable primary health care and public health services across all sectors including housing, increased attention was given to social and environmental determinants of health and health-conducive environments [188,189].

As there is a shift of healthcare services from the hospital to the home due to potential benefits in costs and clinical outcomes, especially for admission avoidance type programmes [190], it is recognised that housing amenities are becoming an important sustainability partnership for healthcare. The partner role of housing can transform mental health services through keeping people with mental health problems independent at good-quality supported housing and enabling timely discharge from hospitals, and thus, improving their quality of life [191]. However, importantly, housing becomes an essential component of the integrated primary health care and public health-based cancer survivorship care model. Expanding the role of delivering different levels of care, including prevention, through housing and indoor environmental conditions is stated at the 2018 Memorandum of Understanding (MoU), signed by several UK government bodies and organisations related to housing, care and health sectors [192]. Within the broad area of public health research, interest in promoting healthy housing and engaging with public health issues is not new [193,194,195]. It is recognised that residents vulnerable to poor domestic conditions experience a higher use of healthcare services and visits to the doctor [196,197]. Although addressing the environmental determinants of health at the housing level is often not under the direct control of primary care, partnership with local providers and architecture and design-related public health practitioners can improve health, while also leading to better use of healthcare resources. Indeed, a British participant sample residing in social housing who were referred to receiving a thermal comfort-related intervention to their homes—installation of new energy efficient boiler and double-glazed windows—resulted in reductions of 4% in anxiety among older residents and of 16% in NHS-use costs over a six-month period [198]. In addition, when the warmth-related dwelling retrofitting targeted British people suffering from a specific chronic medical condition—NHS patients with respiratory diseases—the GP visits were reduced by 28% and 60% after 6- and 18-months of the trial [199]. Thus, housing design and environmental conditions may sustain the viability of health and delivery around certain survivorship care domains through psychosocially supportive spaces that promote recovery. In the context of health effects associated to influencing factors in home settings, pre-COVID-19 systematic and primary research has also shown how the contact with domestic indoor nature [200], view of green spaces [201,202], natural and artificial lighting [203], environment sounds [204], household air pollution [205,206], thermal comfort and building features and ventilation/heating systems following housing improvements [207,208] can impact on residents’ health and mental well-being. Of these primary studies, the majority used non-objective measurements for evaluating the different home environmental domains and health-wellbeing outcomes. The studied home physical setting was typically separate between housing and residential facilities, excluding one from the other in the same review. Moreover, the environmental variables were mainly determined in isolation, which may contribute to missing valuable data around the effects of attenuation/exacerbation of multiple interrelated environmental factors; spaces with residential vegetation can significantly modify the negative effects of air pollutants of the area on the physiological response of mental stress [209]. Importantly, although a necessary exploratory step, the associations tend to establish through cross-sectional and uncontrolled longitudinal analyses, rather than controlled study designs. Therefore, the health impacts of interconnected physical characteristics of housing and less tangible environmental aspects and the causal pathways and clinical questions are poorly understood. Because of the COVID-19 pandemic, new interest in this interrelationship has emerged. For instance, research has explored the spatial dynamics of biotic elements such as Severe Acute Respiratory Syndrome Coronavirus-2 (SARS-CoV-2) highly depend on abiotic environmental parameters in limited indoor spaces, including thermal properties, humidity, solid surfaces/interfaces, indoor air quality, air stability conditions, wind speed and ventilation mechanisms [210,211,212,213]. While much of the primary studies have focused on health and mental health outcomes, less attention has been paid to the measurable physiological response to housing exposure and the interactions between the home environmental parameters and housing design. Fundamental to provide potential housing-based therapeutic intervention strategies is not to overlook the integrative physiology mechanisms underlying mental and physical morbidities and engaging the role of housing into clinical practice.

## 5. The Role of Psychoneuroimmunology, Inflammation and Built Environment on Recovery following Post-Cancer Treatment

### 5.1. PNI Contributions in the Development and Neuroprogression of Mental Disorders

Focused on the multidimensional management of long-term psychological and physical effects and comorbidities among cancer survivors and disease prevention, the PNI framework shapes the integrative care scenario of cancer survivorship with the central role in inflammatory communication network [214,215,216,217]. Cross-communication of immune-neuroendocrine systems and the potential microbiota interplay mediating the humoral responses, the immunomodulation of behaviour and neuroactive signalling, as well as the activation of inflammatory processes translocating signals to distal organs through the circulatory system have gained increased attention [218,219,220]. Inflammatory stimuli initiate a protective and adaptive response to a new stress condition that aims to restore normal tissue architecture through a coordinated resolution programme [221,222]. Across the lifespan, a variety of stimuli—including pathogens, physical inactivity, poor diet, environmental toxicants, psychological stress and disrupted circadian rhythm—sustain this crosstalk machinery to converge on systemic, chronic low-grade inflammation, a shared pathophysiologic response mechanism commonly observed across several mental health conditions and chronic diseases [223,224]. Inflammatory systemic response-derived factors involve tissue-specific immunity [225] and the release of pro-inflammatory mediators by many tissues [226,227]. A complex set of interactions between inflammatory mediators—vasoactive amines and peptides, complement components, lipid mediators, cytokines, chemokines, proteolytic enzymes [228], transcription and growth factors [229], acute-phase proteins [230]—and effector cells—myeloid and lymphoid [231,232], epithelial [233], endothelial [234], stromal [235] cells—as well as their receptors [236], could operate in different step-wise sequences overlapped in chronological terms in getting a pathological neuroimmunological environment to cancer progression [237,238], behavioural symptoms [239] and psychiatric disorders [240].

It is worth highlighting that although microglia activation—resident immune cells of the brain—linked with peripheral inflammation remains unclear [241], it has been recently established that during prolonged peripheral systemic inflammation, the microglia transforms into a phagocytic phenotype associated with inducing the loss of blood–brain barrier integrity to initiate leakage of systemic substances and cause neuroinflammation [242]. Increased evidence supports the immune system as a critical element involved in a subset of neuropsychiatric symptoms in several mental disorders [243].

The lifestyle-based modifiable risk factors and psychosocial factors can influence the microglia activation and the levels of circulating PNI biomarkers and thus, play a key role on the mental disorders [241,244]. Features of the built environment have been systematically associated with those stimuli and consequently with health behaviour changes [245]. For instance, construction or remodelling of new urban infrastructures for walking, cycling and public transportation, street connectivity improvements, parks and playground renovations for active modes of transportation, and accessibility and proximity to these destinations have been related to an increase in physical activity [246,247]. Moreover, spatial exposure to a particular food environment—location, type and number of food outlets—has shown to influence dietary outcomes [248]. These findings have direct implications for planning, designing, and retrofitting health-enhancing residential environments as they could influence residents’ lifestyle behaviour patterns.

Besides inflammageing, the long-lasting effects of past cancer treatment—exposure to chemotherapy and/or radiation—has been associated with increased inflammation markers of accelerated ageing, triggered prematurely in cancer survivors [249,250]. Moreover, the exposure to a persistent, unregulated inflammatory profile contributes to cardiometabolic complications [251,252], other late and long-term adverse effects such as lower cognitive performance [253], fatigue [254,255], depressive disorders and sleep disturbances [256], cancer recurrence [257], and worse survival outcomes in survivors for most common diagnosed cancer types [258,259]. Patients with mental disorders exhibit alterations in the PNI profile with an increase in the levels of peripheral pro-inflammatory cytokines which are thought to underlie many psychiatric conditions [260,261]. Overall, the inflammation and PNI-related indicators have shown significant regulatory response changes in multiple major psychiatric disorders, including major depression, bipolar disorder, obsessive compulsive disorder, sleeping disturbance and suicide [262].

### 5.2. Dual Role of Multi-Level PNI Biomarkers—Clinical Information and Built Environment Exposure

Efforts to identify and translate PNI and inflammation-related factors to a minimally invasive biomarker panel with acute predictive performance are being carried out [263,264]. Over the last decade, efforts have focused on identifying novel panels for larger candidate and score sets that provide more robust diagnostic and prognostic accuracy tools over single tests for both mental disorders, including bipolar disorders, schizophrenia [265,266] and major depressive disorders [267], and cancer [268,269]. Despite the universal expression of mediators in inflammatory response remaining stable in circulation by most tissues [270] and tumour microenvironments [271]—compared with other biomarkers more likely to progressively lose their expression as the disease progresses [272]—there has been no consensus on the methodology used to quantify the added value of biomarker panels.

Systemic and multi-level PNI markers are demonstrated to be useful to capture the clinical and pathologic information, facilitate comprehension into the correlations among various physiological system levels—immune and neuroendocrine [263,264]—and ultimately, to establish interactions between low-grade inflammation physiology of chronic diseases and environmental determinants [273,274].

Low levels of long-term and chronic exposure to ambient air pollution has been shown to have effects on the neuroinflammation and lipoperoxidation involved in the physiopathology of several diseases [275]. Indeed, longitudinal analyses have estimated a two-fold increase of the morbidity from common mental disorders attributable to long-term exposure to residential air pollution [276]. Vulnerable people with pre-existing chronic diseases and living in communities of low socioeconomic position have an increased susceptibility to amplify the acute health effects of air pollution [277]. Thus, patients with co- and multi-morbidities may be more sensitive to certain environmental determinants. Moreover, different comorbidity clusters can differ in terms of systemic, low-grade inflammatory state markers levels [278,279], and specificity of inflammatory biomarkers across clustering of psychiatric disorders exhibits clinical similarity among mental disorders [262]. In newly built houses characterized by sustainable materials and modern building technology (equipped with a modern ventilation system), outdoor sources may have a limited effect on indoor air quality during wintertime. Still, the role of microbes and chemical characterisation of household indoor particles was significantly associated with inflammatory and cytotoxic responses [280]. In this sense, the microbiome of the built environment alters the human microbiome and related immune functions, and has a potential detrimental effect on the mental health of the residents [281].

Regarding disease-specific PNI indicators, human subjects following ambient air pollution exposure report an elevation of secretion of plasma cortisol, and these higher levels has been related to depressogenic cognitive distortions in patients with major depressive disorders [282]. In addition, low serum concentrations of brain-derived neurotropic factor (BDNF) are a state characteristic of depressive disorders [283], and importantly, the exposure to an air pollution-impacted environment may influence changes on serum BDNF concentrations [284].

Biological responses on lighting-induced melatonin concentration have been widely investigated [285]. Architectural spaces with lighting conditions with higher weighted illuminance for intrinsically photosensitive retinal ganglion cells sensitivity can trigger greater suppression of melatonin. This relationship was largely evaluated in a semi-naturalistic environment, which has relevant implications for translating the findings to home environments [286]. Melatonin can modulate the tissue responses boosting the immune system and promote antioxidant defenses for the brain and body. Because of its neuroprotective role, the melatonin as a potential predictive biomarker in ageing and psychiatric disorders is being investigated [287]. Spaces facilitating the nature contact have shown to boost immune function. Long-term exposure to interventions involving indoor horticultural therapy to encourage gardening at home has been associated with reduced markers of inflammageing and immunosenescence in community-dwelling older adults [288]. Furthermore, it has been reported that arduousness of access to and lack of property of outdoor spaces were boundaries to sunlight exposure [289], and dose reaching among participants is dependent on environmental conditions [290].

In addition, dynamic exposure measurements have provided crucial evidence for associating lifestyle supportive built environment exposure to reduced insulin resistance via insulin biomarkers in breast cancer survivors [291], a shared pathogenic mechanism of major depression and cancer [292,293]. This provides crucial evidence for interventions addressing the mental comorbidities and cancer recurrence risk inequities in home environment design of cancer survivors.

### 5.3. Evidence of Intervention Trials and Therapeutic Approaches through Housing Design and Environmental Factors

Systematic research has examined several non-pharmacological intervention strategies for improving the mental and physical effects, comorbidities and survival of cancer via neuro-immune and microbiome mechanisms, including psychosocial therapies [294] and exercise training [295,296,297,298]. In the context of therapeutic spaces [299], under which perceptual and physiological needs should influence environmental solutions, few randomised intervention trials evaluated whether the exposure to household environmental determinants—air quality conditions [300,301,302,303,304,305], daytime lighting and light-at-night [306,307], and nature [308] exposure—can alter characteristics of the immune system and induce stimulation via level changes of one or more systemic PNI biomarker(s).

Table 2 provides the main study characteristics and findings of home modifications-based non-pharmacological interventions for air quality. During active-mode air filtration interventions in home environments, there were reported reductions of between 40% and 60% in particulate matter (PM)_2.5_ concentrations compared with the control trial [300,301,302,303,304,305]_._ The high-efficiency particle air (HEPA) filtration reduced the average household total VOCs concentration by approximately 65% during a one-year period of study follow-up in dwellings of 200 participants [303]. Some air filtration intervention studies did not observe statistically significant effects of exposure to filtered air for certain systemic PNI indicators [300,301,302,303,305]; by contrast, some of these interventions found associations between air filtration and peripheral level changes of interleukin (IL)-6 [301,304], C-reactive protein (CRP) [304,305] and high sensitivity (hs)-CRP [303]. Importantly, the intervention period at home was short-time exposure, comprising of mostly a range between 7-day and 90-day periods, which may unalter the measurable physiological indicators and neuroimmune functions. One intervention trial creating negatively charged-particle dominant-indoor air conditions (NCPDIAC) within homes showed an elevation of endothelial growth factor (EGF), granulocyte-colony stimulating factor (G-CSF) and Eotaxin markers as an immunological response in the individuals exposed to NCPDIAC [309]. These clinical biomarkers may have a neuroprotective and anti-inflammatory effects, and attenuate the neuroinflammation [310,311]. Indeed, the exposure to high-dose air negative ionization has been significantly associated with lower depression severity ratings [312].

Table 3 provides the key characteristics and findings of household artificial lighting and nature environments-based non-pharmacological interventions. One randomised crossover intervention evaluated the exposure to two daytime light epochs at home—280 lx blue-enriched light or 240 lx blue-suppressed light during the morning, and 140 lx and 100 lx blue-suppressed light during the afternoon and evening, respectively—and found no significant difference for salivary melatonin concentration between the two light epochs [307]. However, it has been shown that this routine exposure is associated with increased risk of inflammation [313]. In addition, the short-term exposure to polychromatic lights in the evening suppressed the melatonin secretion; following the intervention, the melatonin levels were decreased for exposures to bathroom and hall daylight white with a high blue component (fluorescent, 130 lx and metal halogenide, 500 lx) [306]. The lamps with a high blue component and high intensities produce more pronounced effects on melatonin concentration. In fact, two-hour exposure to blue-enriched artificial light in the evening showed a suppressive effect for melatonin [285]. These potentially adverse physiological effects could not only be mitigated by appropriate interventions in the spectral distributions and intensities of lighting and housing retrofitting, but also providing domestic spaces to be lived in therapeutically to manage several medical conditions—similar to the bright light therapy in the treatment of patients with mental disorders [314] and mood disorder symptoms in post-treatment cancer survivors [315]. For instance, 30 min/day exposure to the indoor bright light at 1250 lx (~465 nm) in the morning improves fatigue, mood and sleep disorders in cancer survivors [315]. Interestingly, the melatonin neuroendocrine hormone is clinically used as an adjuvant therapy to significantly enhance the overall cancer survival rate, while reducing treatment-related side effects like neurotoxicity and thrombocytopenia [316]. Although further research is necessary, combining bright light and exogenous melatonin therapies on the elderly population has resulted in greater beneficial effects related to sleep disturbance and circadian outcomes than single therapy [317]. With the advancement of gallium nitride Light Emitting Diode (LED) technology, the exposure to ultraviolet B radiation emitting LEDs (293 nm) has shown to be up to two-fold more effective in safely producing vitamin D_3_ in human skin compared to sunlight exposure [318]. The use of LEDs with targeted wavelengths can trigger specific biological responses. As PNI indicator, the vitamin D can potentially modulate and inhibit tumour growth progression by interfering with the inflammation system in the tumour microenvironment [319], and its deficiency has been indeed associated with increased circulating inflammatory biomarkers in cancer stages [320]. Long-term vitamin D supplementation does lead to decreases in both cancer-related mortality once cancer is diagnosed [321] and the development of advanced metastatic cancer [322]. Given that the melatonin and vitamin D could synergistically lead to cytostatic effects—the ability to slow or stop the growth of cells—in cancer cells [323], their combination is emerging as an advantageous therapeutic strategy for quality of life in cancer survivors [324]. These artificial lighting sources could be integrated within cancer survivors’ domestic microenvironments that experience less exposure to sunlight and natural lighting due to housing design features.

The indoor nature-based therapy is also associated with PNI biomarker concentration changes. Particularly, an average of 159-h exposure to an indoor therapeutic garden over a six-month period was associated with a reduction of salivary cortisol levels among older patients with Alzheimer’s Disease with behavioural and psychological symptoms of dementia [308]. Elevated cortisol has been positively correlated with persistent depressive symptoms in a large English population-based cohort of older adults [325]. Nature-based interventions can also engage physical exercise and, ultimately, boost the immune system function, as shown with the horticultural therapy intervention among older people, who reduced T-cell exhaustion and inflammageing, and reduced IL-6 levels, following a six-month period of exposure [288].

In light of this, the psycho-exposome is gaining importance in order to understand the mixture of environment contributors to psychiatric disorders and mental health-related outcomes [326]. Importantly, the exposure to multi-factorial non-pharmacological interventions can reduce the psychotropic medication prescribed to older patients [327]; therefore, non-pharmacological treatments may reduce the need for pharmacological prescriptions. However, home environments are not currently regarded from this perspective at the design stage or building improvement interventions.

## 6. Conclusions

This exploratory review building on the concept of therapeutic architecture is of timely importance, especially when considering the uncertainties deriving from the ongoing COVID-19 pandemic and potential future scenarios of infectious outbreaks. It promotes home environments facilitating recovery and rehabilitation of cancer survivors rather than impeding it, as this population presents unique pathophysiological comorbidity and multimorbidity. Safeguarding the mental health of cancer survivors requires optimal physiological conditions of their dwellings and expanding the role of domestic spaces as environments boosting the neuro-immune function into the standard of cancer survivorship supportive and preventive care. A PNI-oriented approach may offer translational opportunities for integrating mental and physical health care through home environments and provide a unique care scenario which impacts on transforming the healthcare system, primary care, public health and housing facility design for cancer survivorship.

This paper provides evidence that interventions addressing housing improvements and clinical implications by using PNI outcomes might be effective, non-pharmaceutical treatments to promote physiological health and to treat cancer survivors’ mental health care. While the effects of several household environmental factors and architectural aspects on self-reported measures of mental health have been well-investigated, the evidence of the effects on the PNI system had not been approached so far. Despite the advantages of this narrative review exploring the environmental parameters under a real-world exposure scenario in naturalistic home settings, future research is imperative to evaluate the long-term effects. Future interventions could expand the PNI indicators, using a comprehensive panel of biomarker combinations linked to risk exposures for monitoring the individual’s inflammation status and PNI profile. Intervention trial designs developed by primary care staff, mental health professionals and building scientists should begin to critically evaluate this new approach. Eventually, such findings should link policies for enabling environment arguments to cancer survivorship frameworks.

Ultimately, this exploratory review serves as a precursor for a subsequent comprehensive systematic review and meta-analysis and provide a first strategic framework consolidating information about several factors that should be considered when developing it in order to maximise the research impact (Figure 1). Addressing this treatment approach may gather evidence to link housing context–PNI mechanism–mental health/physical outcome configurations to residential built environment design.

## Figures and Tables

**Figure 1 ijerph-18-10891-f001:**
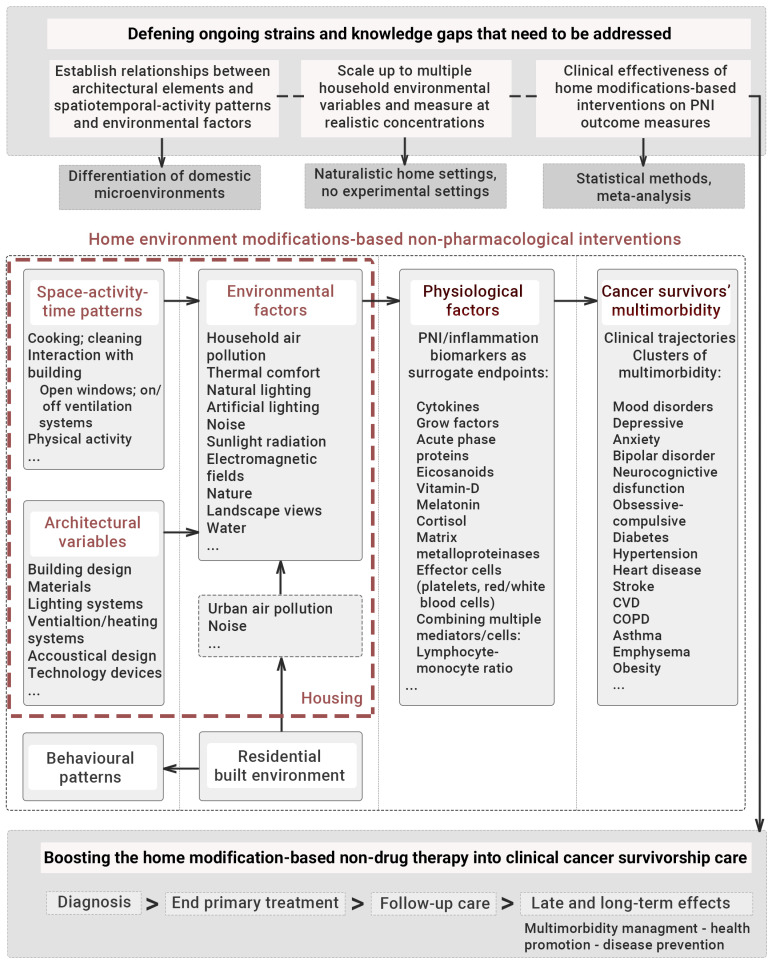
Initial framework outlining the core factors for completing a comprehensive systematic study in non-pharmacological interventions based on home environment modifications.

**Table 2 ijerph-18-10891-t002:** Summary of key characteristics and findings of studies addressing the associations between household air quality-based non-pharmacological interventions and PNI biomarkers.

First Author (Date)	Study Location	Study Design (Period)	Environmental Factor	Intervention	PNI Biomarkers ^1^	Key Findings
Air quality-based interventions						
Karottki (2013) [300]	Copenhagen, DK	Randomised, double-blind, crossover intervention (November 2010–June 2011)	PM_2.5_; PNC; PAH	Recirculated particle-filtered or sham-filtered indoor air (HEPA filter)/14-day period for each exposure. Time home median = 83%	Haemoglobin, leukocyte counts, lymphocytes, monocytes, CRP, HDL, LDL	Reductions in the median concentration of: PM_2.5_, from 8 to 4 μg/m^3^; PNC, from 7669 to 5352 particles/cm^3^; PAH, from 0.5 to 0.25 ng/m^3^.
						No statistically significant effects on inflammatory biomarkers.
Padró-Martínez (2015) [301]	Somerville, MA, USA	Randomised, double-blind, crossover trial (February 2011–November 2012)	PNC	Recirculated particle-filtered or sham-filtered indoor air (HEPA filter)/21-day period for each exposure. Time home median = 81%	hs-CRP, fibrinogen, TNF-RII, IL-6	Reductions in the median concentration of PNC, from 6820 to 4850 particles/cm^3^.
						IL-6 concentrations were significantly higher following HEPA filtration (0.668 pg/mL; CI = 0.465–0.959).
Shao (2017) [302]	Beijing, CN	Randomised, double-blind, crossover trial (December 2013–March 2014)	PM_2.5_; BC	Active-mode and sham-mode air filtration (HEPA filter)/14-day period for each exposure. Time home median = 95%	CRP, fibrinogen, IL-6, IL-8	Reductions in the median concentration of: PM_2.5_, from 60 to 24 μg/m^3^; BC, from 3.9 to 1.8 μg/m^3^.
						Significant reductions in IL-8, from 120.30 to 47.65 pg/mL.
Chuang (2017) [303]	Taipei, CN	Randomised, double-blind, crossover trial (January 2013–December 2014)	PM ≤ 2.5 μm; total VOCs	Active-filtered indoor air and placebo air conditioner filter/365-day period for each exposure	hs-CRP, fibrinogen	Reductions in the median concentration of: PM_2.5_, from 21.4 to 12.8 μg/m^3^; VOCs, from 1.2 to 0.43 ppm.
						Significant reduction in hs-CRP, from 0.36 to 0.18 mg/dL.
Allen (2011) [304]	Smithers, British Columbia, CA	Randomised, crossover trial (November 2008–December 2009)	PM2.5 mass concentration; LG, a marker of WS PM	Active-mode and sham-mode air filtration (HEPA filter)/7-day period for each exposure. Time home median = 77%	CRP, IL-6	Reductions in the median concentration of: PM_2.5_, from 11.2 to 4.6 μg/m^3^; LG, from 127 to 33 ng/m^3^.
						Decreased concentrations of CRP (32%) and IL-6 following active-mode air filtration.
Kajbafza-deh (2015) [305]	Vancouver, British Columbia, CA	Randomised, single-blind, cross-over intervention (December 2011–August 2012)	PM_2.5_ mass concentration; LG, a marker of TRAP PM	HEPA filtration and placebo filtration/7-day period for each exposure. Time home median = 74%	CRP, IL-6	Reductions in PM_2.5_, from 7.1 to 4.3 μg/m^3^ following HEPA filtration.
						Increase in CRP levels with increasing indoor PM_2.5_ concentrations.
Nishimura (2015) [309]	Japan	Non-randomised, non-blinded trial (21 months)	NCPDIAC; negatively charged- particles	NCPDIAC and indoor conditions with device generating NCP OFF/3-month period for each exposure	Measures of 29 PNI markers ^2^	EGF, G-CSF and Eotaxin concentrations increased during NCPDIAC trials.

BC, black carbon; BLC, blue component; CA, Canada; CN, China; CRP, C-reactive protein; DK, Denmark; EGF, endothelial growth factor; G-CSF, granulocyte-colony stimulating factor; HDL, high-density lipoprotein; HEPA, high-efficiency particle air; hs-CRP, high sensitivity C-reactive protein; IL, interleukin; LDL, low-density lipoprotein; LG, levoglucosan; MA, Massachusetts; NCPDIAC, negatively charged-particle dominant indoor-air conditions; PAH, polycyclic aromatic hydrocarbons; PM, particulate matter; PNC, particle number concentration; PNI, psychoneuroimmunology; TNF-RII, tumor necrosis factor receptor II; TRAP, traffic-related air pollution; VOCs, volatile organic compounds; WS, woodsmoke. ^1^ Blood samples were collected from participants for measuring the systemic psychoneuroimmunology biomarkers. ^2^ EGF; eotaxin/CCL11, C-C motif chemokine ligand 11; VEGF, vascular endothelial growth factor; TNF-β; TNF-α; MIP-1β, macrophage inflammatory protein 1-beta; MIP-1α, macrophage inflammatory protein 1-alpha; MCP-1, monocyte chemoattractant protein; IP-10, interferon-inducible protein 10; IL-17 A; IL-15; IL-13; IL-12 (p70); IL-12 (p40); IL-10; IL-8; IL-7; IL-6; IL-5; IL-4; IL-3; IL-2; IL-1ra; IL-1β; IL-1α; interferon (IFN)-γ; IFN-α2; GM-CSF, granulocyte-macrophage colony stimulating factor; G-CSF.

**Table 3 ijerph-18-10891-t003:** Summary of key characteristics and findings of studies addressing the associations between domestic lighting and nature-based non-pharmacological interventions and PNI biomarkers.

First Author (Date)	Study Location	Study Design (Period)	Environmental Factor	Intervention	PNI Biomarkers ^1^	Key Findings
Artificial lighting-based interventions						
Wahn-schaffe (2013) [306]	Berlin, Germany	Randomised, crossover design (February 2007)	Artificial lighting; LAN exposure	Exposure to polychromatic lights via everyday domestic lamps of different types in two intensities (130 and 500 lx at the cornea) and spectral distributions in the evening/6-day period of exposure	Melatonin	Melatonin levels decreased from 44 ± 26 at baseline to 23 ± 9 (bathroom daylight white), and 22 ± 26 (hall daylight white). Exposure to lighting conditions in the evening reduced melatonin levels both during and after the intervention, even the light with the lowest blue proportion. Yellow light showed no effect on melatonin concentrations.
Sander (2015) [307]	Albertslund, Copenhagen, DK	Randomised cross-over design (October 2013–November 2013)	Artificial lighting; day-time light	Exposure to two light epochs: blue-enriched light and blue-suppressed light conditions from 8 am to 1 pm/21-day period for each exposure. After 1 pm, 140 lx blue-depressed light in both epochs; 6 pm-bedtime, 100 lx blue-suppressed light	Melatonin	No significant difference between blue-enriched and blue-suppressed light for saliva melatonin concentrations.
Nature-based interventions						
Pedrinolla (2019) [308]	Mantua, Italy	Single-blind randomized controlled trial (June 2015–2016)	Indoor natural environment	Exposure to indoor therapeutic garden (experimental group) or standard care environment (control group)/sessions of 2 h, 5 times/week for 6-month period	Cortisol	Significant reductions in cortisol levels, from −6.4 to −2.1 Nmol/L.

DK, Denmark; LAN, light at night; PNI, psychoneuroimmunology. ^1^ Salivary samples were collected from participants for measuring the systemic psychoneuroimmunology biomarkers.

## Data Availability

No new data were created or analyzed in this study. Data sharing is not applicable to this article.

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
