# Peer review of "The Interplay between Housing Environmental Attributes and Design Exposures and Psychoneuroimmunology Profile—An Exploratory Review and Analysis Paper in the Cancer Survivors’ Mental Health Morbidity Context"

_ijerph, 2021, doi:10.3390/ijerph182010891_

Round 1

Reviewer 1 Report

I thank the authors for this review of the effect of housing environment and design factors on cancer survivors’ mental health and the potential implications in the non-pharmacological treatment in their living spaces.

In general, this review has been well organized with several parts and the achieved findings could be interesting. Several aspects could be considered to improve this paper:

Section 3: Several environmental factors have been discussed, including air qualities and ventilation, green space /view / light, and noise and indoor acoustic performance. 1) However, it might be not clear why the thermal environment (thermal comfort issues) has not been toughed? 2) In addition, indoor lighting (natural / artificial and light colour) could be a critical issue affecting human biological performance, especially for the cancer survivors staying at home. Is it possible to enhance this part?

Presentation: The structure of this paper is general ok. It might be useful to consider applying Tables to compare the key literatures and summerize the key points. For example, section 3.1, 3.2, 3.3, and section 4.2 and 5.3. This way could help enhance the readability of this paper.

Reviewer 2 Report

The article is potentially important to the field, providing a review and analysis of previous work in psychoneuroimmunology and therapeutic architecture. The article is well organised. As a large number of literature (>300) was considered in this review, the following suggestions are given to the authors:

1. Researchers in oncology, psychoneuroimmunology, mental health, therapeutic architecture and others may find this article interesting and relevant to their work. It is inevitable to use jargons to present ideas precisely, however, readers from different fields may the article difficult to understand. The authors are suggested to make more explicitly the findings/insights gained from the review.

2. In Conclusions, the authors argued that this review “… serves as a precursor for a subsequent comprehensive systematic review and provide a first strategic framework consolidating information about several factors that should be considered …” (lines 668-670). Yes, the factors may now be scattered in the paragraphs, but what are the factors in summary and what the prototype framework looks like?
